# Physical Activity as a Regulatory Variable between Adolescents’ Motivational Processes and Satisfaction with Life

**DOI:** 10.3390/ijerph16152765

**Published:** 2019-08-02

**Authors:** Mikel Vaquero Solís, Pedro Antonio Sánchez-Miguel, Miguel Ángel Tapia Serrano, Juan J. Pulido, Damián Iglesias Gallego

**Affiliations:** 1Teacher Training College, University of Extremadura, Avd. de la Universidad S/N, 10003 Cáceres, Spain; 2Faculty of Sport Science, University of Extremadura, Avd. de la Universidad S/N, 10003 Cáceres, Spain; 3Faculty of Sport Science, Faculty of Movement Sciences, University of Lisbon, 1499-002 Lisboa, Portugal

**Keywords:** motivation, physical activity, well-being, adolescents, school health

## Abstract

Framed within Self-Determination Theory, the objective of this study was to analyze the relationship between satisfaction and frustration of basic psychological needs, levels of motivation, physical activity, and satisfaction with life. Methods: A total of 487 students participated, comprising males (*n* = 262) and females (*n* = 225), aged between 14 and 16 years (*M* = 15.02; *SD* = 0.87), from different secondary schools. Results: A regression analysis was carried out (structural equation modeling) that revealed the existence of two theoretical lines, one positive and the other negative, where the satisfaction of basic psychological needs was positively related to autonomous motivation and physical activity, which predicted satisfaction with life. On the other hand, the frustration of basic psychological needs was positively related to controlled motivation, whereas controlled motivation (introjected regulation and extrinsic regulation) was inversely associated with physical activity and satisfaction with life. Conclusion: The results show the importance of motivational processes in physical activity, and the effects of physical activity on satisfaction with life in adolescents who spend more time engaged in physical activity.

## 1. Introduction

Adolescence is characterized by being a period in which young people cope with the difficult task of forming their personality and identity and establishing their relational network [1]. In this regard, sports experiences and physical activity levels in adolescents both play an important role in their personal growth and development in physical, cognitive, social, and moral terms [2,3]. Based on this, satisfaction with life is a key indicator of mental health and subjective well-being [4], which refers to how people evaluate their lives both in general and in specific domains such as family, friends, and leisure time [5,6]. Diener [7] pointed out three components of well-being: positive affect, negative affect, and satisfaction with life. This theory was subsequently extended, with happiness as the fourth component [8]. Previous studies have pointed out the importance of satisfaction with life for adolescents’ adequate growth and development, favoring the improvement of social relationships and preventing depressive symptoms, stress, and anxiety [9,10].

The concept of satisfaction with life has been related to other factors of psychological well-being such as self-esteem [11], stress [12], emotional processes [13], academic performance [14], social climate, and physical activity [15]. In relation to physical activity, it should be noted that from the perspective of mental well-being [16], psychological well-being refers to optimal psychological functioning [17]. Hence, physical activity plays a fundamental role in the maintenance of vital functions and a healthy lifestyle [18]. Most studies conducted with adolescents show that physical activity is associated with greater satisfaction with life [9,19,20]. Drawing on this evidence showing that physical activity is related to satisfaction with life [21], some investigations have addressed the mechanisms that facilitate participation in physical activities and enhance well-being [22,23]. 

On the other hand, satisfaction with life has been related to motivational processes [24,25]. Along these lines, a meta-analysis developed by the authors of [26] suggested that Self-Determination Theory [27,28] constitutes a possible frame of reference for understanding healthy behaviors and motivational processes related to health and well-being. Likewise, it is important to highlight previous studies that have focused on the role of basic psychological needs in subjective well-being [29]. In this regard, children whose basic psychological needs are not met are more prone to emotional imbalance [30] and antisocial behaviors if the need for relatedness is not fulfilled [31]. On the contrary, the satisfaction of autonomy and competition positively predicts voluntary participation [32].

Self-Determination Theory focuses on human motivation, behavior, and personal well-being [33,34]. Thus, Deci and Ryan [27,28] stated that behavior can be divided into three stages according to higher or lower levels of self-determination: autonomous motivation, controlled motivation, and amotivation [35]. Autonomous motivation (intrinsic regulation, integrated regulation, identified regulation), is characterized by a sense of personal will, where the behavior and the activity carried out are meaningful. This occurs when a person engages in a behavior because it is perceived as being consistent with the person’s own intrinsic goals and it is felt to be important or interesting. On the other hand, controlled motivation (introjected regulation, external regulation), represents behavior that emerges from feelings of pressure, punishments, feelings of shame, external rewards, or approval. Finally, amotivation is a condition that reflects a lack of positive attitude and a feeling of uselessness, and the absence of autonomous or controlled motivation that is required to persist in an activity.

Moreover, Deci and Ryan [35] added that the central distinction within contemporary work on Self-Determination Theory is autonomous motivation (intrinsic, integrated, and identified motivation) versus controlled motivation (introjected and external regulation). In this sense, autonomous motivation is characterized by a sense of personal will, where the behavior and the activity carried out are meaningful. This occurs when a person engages in a behavior because it is perceived as consistent with the person’s own intrinsic goals and it is felt to be important or interesting [36]. Controlled motivation represents behavior that emerges from feelings of pressure, punishments, feelings of shame, external rewards, or approval [35]. Finally, amotivation refers to a lack of positive attitude or a feeling of uselessness.

Research based on Self-Determination Theory has already assessed the mediating value of motivation in relation to basic psychological needs [37], quality of life [38], physical activity [39], and subjective well-being [40]. According to this theory, the type of motivation shown by people will depend on the satisfaction, to a greater or lesser degree, of their basic psychological needs— competence, autonomy, and relatedness—which, in turn, will determine individuals’ level of well-being [41,42]. The need for competence is characterized by the feeling of effectiveness while carrying out an optimally challenging task. Autonomy refers to the initiative felt by people when participating voluntarily in the proposed activities, and finally, the need for relatedness refers to the need to maintain satisfactory social relationships and to feel emotionally connected to others [42]. 

Based on the foregoing, previous studies have already addressed the effects that motivational processes related to the practice of physical activity can have on the markers of well-being, such as satisfaction with life [20,24,25,43,44], although, other studies showed a negative relationship [45,46]. However, previous studies had a series of limitations. In some cases they did not value physical activity since the research took place in a sports or school context (physical education) [20,43,45].Moreover, some did not use a complete conceptualization of Self-Determination Theory [24,25], or they were based on a different conceptual framework for testing satisfaction with life [20]. This study is different from others of its nature because it deals with the extent to which the satisfaction/frustration of basic psychological needs affects adolescents’ motivation to engage in physical activity, and the effect of physical activity on their satisfaction with life. Therefore, our objective was to determine the degree to which motivation, along with basic psychological needs and physical activity, predicts satisfaction with life. Therefore, it was postulated that satisfaction of basic psychological needs would be positively related to autonomous motivation, physical activity, and satisfaction with life. Likewise, the second hypothesis was that frustration of basic psychological needs would be negatively related to controlled motivation and satisfaction with life.

## 2. Materials and Methods 

### 2.1. Participants

A total of 487 Spanish secondary school students aged between 14 and 16 years (*M* = 15.02, *SD* = 0.87), including 262 (53.8%) males and 225 (46.2%) females, participated in this investigation. They were from five different schools in the region of Extremadura (Spain). The sample was selected through hierarchical cluster sampling, considering the distance of the schools, whether the teaching staff were known, and the availability and the time required for the researcher to travel.

### 2.2. Instruments

Satisfaction of Basic Psychological Needs. To assess the satisfaction of basic psychological needs, the Spanish version [46] of the Basic Psychological Needs in Exercise Scale (BPNES) [47] was used. This questionnaire is composed of 12 items grouped into three factors, which begin with the initial stem phrase, “*In my Physical Education classes*…” followed by the items. The factors are: Autonomy Satisfaction (four items, e.g., “*I have the opportunity to choose how to perform the exercises”*, α = 0.74), Competence Satisfaction (four items, e.g., “*I think I can meet the demands of the class”*, α = 0.69), and Relatedness Satisfaction (four items, e.g., “*I feel that I can communicate openly with my classmates*”, α = 0.80). Items were rated on a 5-point Likert scale, ranging from 1 (*completely disagree*) to 5 (*completely agree*).

Frustration of Basic Psychological Needs. To assess basic psychological needs frustration, the Spanish version (EFNP) [48] of the Psychological Need Frustration Scale (PNTS) [49] was used. The instrument is composed of 12 items grouped into three factors, which begin with the initial stem phrase, “*In my physical exercise…*” followed by the items. The factors are: Autonomy Frustration (four items, e.g., “*I feel pressured to behave in a certain way*”, α = 78), Competence Frustration (four items, e.g., “*There are situations where I feel incapable*”, α = 79), and Relatedness Frustration (four items, e.g., “*I feel as if other people do not like me*”, α = 77). Items were rated on a 5-point Likert scale, ranging from 1 (*completely disagree*) to 5 (*completely agree*).

Motivation towards physical activity. To assess motivational regulation towards physical exercise, the Behavior Regulation Questionnaire in Exercise, [50] and adapted and validated in Spanish, [51] was used. This instrument is composed of 23 items grouped into six factors: Intrinsic Regulation (four items, e.g., “*because I think exercise is fun*”, α = 88), Integrated Regulation (four items, e.g., “*because it matches my way of life*”, α = 89), Identified Regulation (three items, e.g., “*because I value the benefits of physical exercise*”, α = 80), Introjected Regulation (four items, e.g., “*because I feel guilty when I do not practice it*”, α = 70), Extrinsic Regulation (four items, e.g., “*because others tell me that I should do it*”, α = 70), and Amotivation (four items, e.g., “*I do not see why I have to do it*”, α = 74), all introduced by the initial stem phrase, “*I do physical exercise*...”. The responses to the questionnaire were rated on a Likert scale ranging from 1 (*not at all true*) to 5 (*completely true*). 

Perception of Physical Activity Levels. Physical activity was analyzed with the Physical Activity Questionnaire for Adolescents [52] adapted and validated in Spanish. [53] This questionnaire is composed of nine items that assess the level of physical activity that the adolescent performed in the last seven days, using a 5-point Likert scale: during his/her free time, during Physical Education classes, as well as at different times during school days (lunch, afternoon, and evening) and during the weekend. The result is a score ranging from 1 to 5 that represents graduated levels of physical activity. The Cronbach’s alpha coefficient obtained was 0.81.

Satisfaction with life. The Life Satisfaction Scale (SWLS) [54] was adapted and validated in Spanish. [55] The instrument is composed of five items that evaluate satisfaction with life through participants’ global judgment. Responses are rated on a 5-point Likert scale ranging from 1 (*totally disagree*) to 5 (*totally agree*). The Cronbach alpha was α = 0.82. 

### 2.3. Procedure

This study was implemented in several phases. Firstly, collaborating schools were contacted to request the pertinent permissions. We explained that participation was voluntary and anonymous, so participants’ identities were not compromised. The study was previously approved by the ethics committee of the University of Extremadura, with the code “89/2016”. All participants were treated in accordance with the ethical principles and codes of conduct of the American Psychological Association [56] for these kinds of investigations. Permission was requested from teachers and parents, and we explained the purpose of the study and which variables would be evaluated. Later, an appointment was made to administer the questionnaires to groups of students in the classroom. Before distributing the questionnaires, we explained the general purpose of the study to the participants, making it clear that it was not an evaluation test, in order to obtain sincere responses. The questionnaires were completed in approximately 25 minutes.

### 2.4. Data Analysis

The statistical package SPSS 23.0 was used for data analysis, performing different tests to determine the nature of the data; the Kolmogorov-Smirnov test for independent samples, the Rachas random test, and the homoscedasticity or Levene’s test for equality of variances. Subsequently, descriptive statistics and bivariate correlations were analyzed for all study variables. 

The statistical package Mplus 7.0 [57] was also used to verify the predictive capacity (structural equation model, SEM) of satisfaction and frustration of basic psychological needs, autonomous motivation, and physical activity, for satisfaction with life.

To test the indirect effects, the path model was re-estimated using bootstrapping resampling procedures (*N* = 5000) to compute 95% bias-corrected confidence Intervals (95% BcCI) [58]. If the 95% BcCI did not include zero, the indirect association was deemed significant. This model was estimated using the maximum likelihood (ML) estimator since MLR with bootstrapping is not yet available in Mplus.

## 3. Results

### 3.1. Preliminary Analysis

Table 1 shows the results obtained in the confirmatory factor analysis (CFA) of each of the scales used. First, CFA was performed on the physical activity questionnaire, which confirmed the presence of a latent variable. Both the BPNES and the EFNP had a structure of one first-order factor consisting of 12 indicators and three fixed factors. In the Behavioral Regulation in Exercise Questionnaire-3 (BREQ-3), a confirmatory factor analysis was performed, with six first-order variables (intrinsic regulation, integrated regulation, identified regulation, introjected regulation, external regulation, and demotivation) that predicted two second-order variables (autonomous motivation and controlled motivation). Finally, CFA was also carried out including the five items of satisfaction with life. As can be seen in Table 1, the comparative fit index (CFI) and the Tucker-Lewis index (TLI) showed values higher than 0.90 [59]. In addition, scores between 0.02 and 0.07 were found in the standardized root mean square residual (SRMR) and the root mean square error of approximation (RMSEA), respectively. In this context, values lower than 0.08 for SRMR and RMSEA are satisfactory [60]. 

Table 2 shows the descriptive statistics for the study variables. The values obtained for Satisfaction of Social Relations and Identified Regulation presented the highest mean values. On the other hand, the frustration variables of the basic psychological needs obtained the lowest mean scores. Likewise, asymmetry and kurtosis analyses reflected moderate asymmetric leptokurtic values.

Table 2 also depicts the bivariate correlations. Each of the basic psychological needs shows a positive relationship (*p* < 0.01) with the levels of Autonomous Motivation, Physical Activity and Satisfaction with Life. In relation to the associations that existed between the frustration of basic psychological needs and the remainder of the variables, the analysis of correlations showed negative relationships (*p* < 0.01) between frustration and satisfaction of basic psychological needs, levels of Autonomous Motivation, and Satisfaction with Life. Physical activity was positively and significantly related (*p* < 0.05) to the satisfaction of basic psychological needs, levels of Autonomous Motivation, and Satisfaction with Life. On the other hand, Satisfaction with Life was not related to frustration of basic psychological needs. Finally, Controlled Motivation was positively associated with physical activity.

### 3.2. Structural Equation Model

A model with the following structure is presented (Figure 1). The measurement model consisted of four latent variables. Firstly, to construct the model, satisfaction of the basic psychological needs was composed of the variables of competence satisfaction, autonomy satisfaction, and relatedness satisfaction. The Frustration of Basic Psychological Needs was created from the scores in Autonomy Frustration, Competence Frustration, and Relatedness Frustration. Autonomous motivation was created by grouping the most self-determined levels of motivation (intrinsic regulation, integrated regulation and identified regulation), and the Physical Activity score was made up of the items of the Physical Activity Questionnaire for Adolescents (PAQ-A).

The initial model showed the following fit indices, which were not acceptable: MRLχ^2^ = 280.052, *p* < 0.05, CFI = 0.87, TLI = 0.93, SRMR = 0.095, and RMSA = 0.079. Therefore, we decided to restructure the model, following the principles of Self-Determination Theory. This model was completed with the formation of two more latent variables, Autonomous Motivation and Controlled Motivation [26,35,41], which were made up of intrinsic regulation, and integrated and identified motivation for Autonomous and External regulation, and introjected regulation for Controlled Motivation. This new model (Figure 2), offered the following acceptable values: MLRχ^2^ = 419.964, *p* < 0.05, CFI = 0.91, TLI = 0.90, SRMR = 0.083, and RMSA = 0.068. 

Subsequently, from the values obtained for Model 2, the model’s structural invariance as a function of gender was analyzed. In this regard, the data presented better fit indices for females (MLRχ^2^ = 354.214, *p* < 0.05, CFI = 0.90, TLI = 0.89, SRMR = 0.095, and RMSA = 0.071), than for males (MLRχ^2^ = 276.033, *p* < 0.05, CFI = 0.88, TLI = 0.87, SRMR = 0.087, and RMSA = 0.081), with the main differences shown in the physical activity–satisfaction with life relationship, which was more notable in males than in females.

The indirect effects between the variables showed firstly that need satisfaction was positively and significantly related (*p* = 0.016) to physical activity (PA) levels via types of motivations (β = 0.285, 95% BcCI = 0.090, 0.480), whereas need frustration was not significantly (*p* > 0.05) associated with PA levels via types of motivation (β = 0.088, 95% BcCI = −0.104, 0.281). Secondly, need satisfaction was positively but not significantly (*p* > 0.05) related to satisfaction with life (β = 0.027, 95% BcCI = −0.003, 0.057), via types of motivation and PA levels. In addition, need frustration was not also significantly (*p* > 0.05) associated with satisfaction with life (β = 0.008, 95% BcCI = −0.013, 0.029) via types of motivation and PA levels. Finally, autonomous motivation was significant indirectly (*p* = 0.03) related to satisfaction with life via PA levels (β = 0.057, 95% BcCI = 0.013, 0.101), and controlled motivation was not significantly (*p* > 0.05) associated with satisfaction with life (β = 0.008, 95% BcCI = −0.002, 0.018).

## 4. Discussion

Grounded in Self-Determination Theory [27,28], the purpose of this research was to analyze the relationships between satisfaction with life, levels of motivation, basic psychological needs, and physical activity. In addition, it assessed the degree to which the structural equation model presented predicted satisfaction with life in adolescents.

Regarding the first hypothesis, which referred to the existing relationships between the levels of autonomous motivation (intrinsic regulation, integrated regulation, and identified regulation), satisfaction of the basic psychological needs, physical activity levels, and satisfaction with life, the present study showed that the levels of autonomous motivation were associated with satisfaction of basic psychological needs. This confirms the importance of these variables in the motivation to engage in physical activity for its mere enjoyment [41]. Previous studies also confirmed this positive association between the levels of more autonomous motivation and the satisfaction of basic psychological needs in many other domains (e.g., school, sport, and work) [28,61]. Satisfaction of the basic psychological needs and autonomous motivation were also positively associated with physical activity and satisfaction with life [41,42]. Previous studies also confirmed this relationship [25], stating that students with more self-determined motivation were more likely to have their basic psychological needs met and therefore, they presented greater well-being or satisfaction with life. In this regard, Lubans et al. (2015) [62] pointed out that changes in basic psychological needs and autonomous motivation were associated with well-being. Therefore, people who feel competent at the task they perform within their peer group will present greater satisfaction with life in all its aspects. Regarding the relationship between physical activity and autonomous motivation, it was stated in [39], that people with autonomous motives were more likely to engage in activities that required some effort. 

In this sense, it was emphasized in [63] that autonomous motivation for adolescents should be fostered through support for autonomy, offering alternative activities that have a rational meaning (autonomy), supporting their confidence and skills, recognizing their efforts (competence), and encouraging positive interaction with their peers (relatedness). Thus, previous studies developed in the school context showed the influence of the motivational climate of the physical education teacher when developing healthy living habits [64,65]. According to Self-Determination Theory (SDT), people can experience deep well-being when their social environment supports their innate needs for competence, relatedness and, above all, autonomy. Furthermore, our findings showed a positive relationship between physical activity and satisfaction with life. These results are consistent with previous studies [9,15,20]. In this regard the authors of [19], in their longitudinal study of physical activity and satisfaction with life, showed that sports had a positive effect on satisfaction with life, and that physical activity had a greater impact on lifetime satisfaction with physical activity for women. In contrast to these gender differences, our study showed that the positive relationship was not applicable to both genders, as physical activity was a more powerful determinant of satisfaction with life for males than for females. This may be influenced by other variables, such as practice time, nature (moderate or vigorous), and the type of physical activity (individual or group), or psychosocial antecedents [66]. In this sense, previous studies indicated the variable active leisure time was a precursor to satisfaction with life [67,68]. 

However, as postulated in the initial hypothesis, our results showed significant negative relationships between satisfaction with life, physical activity, and external regulation. Although the results confirmed a negative relationship, this was not significant for introjected regulation. Similar results were obtained in [44], in which controlled motivation was correlated negatively, but not significantly, with physical activity and satisfaction with life. Also, controlled motivation was positively and significantly related to frustration of basic psychological needs. In this regard, it was suggested in [45] that controlled motivation positively predicted the frustration of basic psychological needs.

As shown in the results obtained, this study has emphasized satisfaction with life as an objective means for assessing well-being. Likewise, the model presented has shown how the satisfaction of basic psychological needs predicted people’s greater autonomous motivation, which translates into more physical activity and, therefore, greater satisfaction with life. In addition, the study revealed that satisfaction with life was predicted by physical activity in males, but not in females. 

### Limitations and Final Remarks

Although this research has contributed new knowledge based on the Theory of Self-Determination, the cross-sectional nature of the study is considered a limitation which has shown that this theoretical model may vary depending on the behavior of each group of students. Therefore, it may also be influenced by the teacher and by other agents of the surrounding setting, such as family and friends, and the socioeconomic environment. Another limitation is the non-inclusion of an objective instrument for the assessment of physical activity levels. Quasi-experimental investigations with repeated measures should be performed to reveal the degree to which each agent affects the motivation toward physical activity and, consequently, its improvement in the sense of well-being. Students who actively participate in active leisure activities will thereby improve their physical and mental well-being.

Thus, the creation of an adequate motivational structure in the teaching staff is an important aspect to consider within the school. In this sense, a teacher trained in the satisfaction of basic psychological needs will better meet the students’ needs for autonomy, competence, and relatedness; thus, creating an ideal learning climate in the classroom. This is consistent with the concept of “education for everyone” and promotes physical education according to the needs, capacities and limitations of each student, thereby favoring their integral education. Likewise, this satisfaction of basic psychological needs will be translated into greater motivation when facing the activities or challenges proposed by the teacher, improving levels of physical activity and, therefore, increasing well-being. Some of the proposed implications can be applied both to the teaching style, such as making classes more personalized, and to improving the students’ well-being, making them feel more integrated and satisfied. In addition, this motivational structure can also be introduced to teachers in the school, creating a motivated and satisfied staff who take more interest in their students’ learning. The well-being of teachers and students at school is of great importance, and this well-being is the reflection of a high-quality education.

## Figures and Tables

**Figure 1 ijerph-16-02765-f001:**
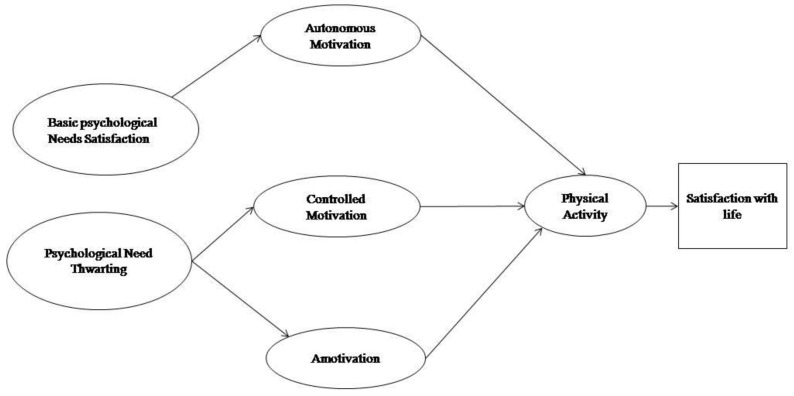
Hypothesized model in adolescents.

**Figure 2 ijerph-16-02765-f002:**
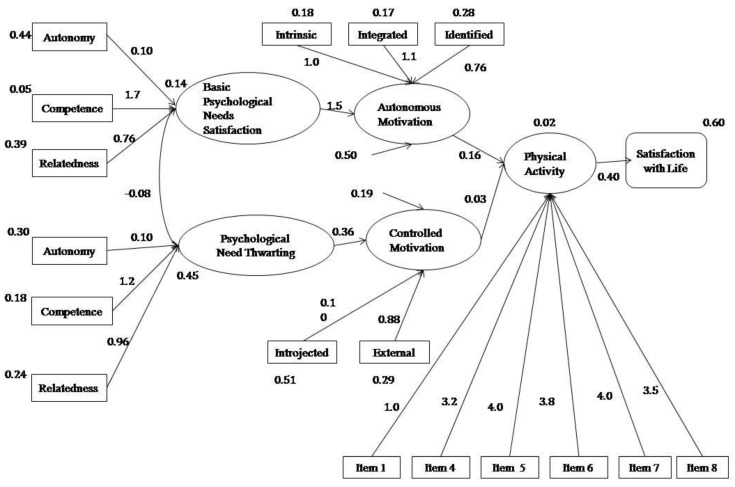
Structural equation model in adolescents.

**Table 1 ijerph-16-02765-t001:** Confirmatory factor analysis of all the scales.

Variables	CFI	TLI	SRMR	RMSEA
Physical Activity	0.97	0.95	0.03	0.06
Basic Psychological Needs Satisfaction	1.00	1.00	0.00	0.00
Psychological Need Frustration	1.00	1.00	0.00	0.00
Autonomous Motivation	1.00	1.00	0.00	0.00
Controlled Motivation	0.97	1.00	0.00	0.00
Satisfaction with Life	0.97	0.95	0.02	0.07

Confirmatory factor analysis of the all scales included in the model.

**Table 2 ijerph-16-02765-t002:** Descriptive analysis and bivariate correlations of the variables.

Variables	1	2	3	4	5	6	7	8	9	10	11	12	13	14	15
1. C.S	-	0.46 **	0.38 **	−0.32 **	−0.18 **	−0.16 **	0.52 **	0.58 **	0.42 **	0.06	−0.21 **	0.37 **	0.32 **	−0.07	0.55 **
2 A.S	-	-	0.29 **	−0.10 **	−0.14 **	−0.04	0.27 **	0.25 **	0.17 **	0.10 *	0.02	0.18 **	0.14 **	0.08	0.25 **
3. R.S	-	-	-	−0.28 **	−0.32 **	−0.34 **	0.20 **	0.20 **	0.20 **	−0.01	−0.11 *	0.08	0.22 **	−0.07	0.22 **
4. T.C	-	-	-	-	0.69 **	0.70 **	−0.12 **	−0.17 **	−0.09 *	0.24 **	0.27 **	−0.08	−0.26 **	0.31 **	−0.14 **
5. T.A	-	-	-	-	-	0.66 **	−0.07	−0.07	−0.05	0.24 **	0.28 **	−0.04	−0.24 **	0.32 **	−0.07
6. T.R	-	-	-	-	-	-	−0.08	−0.06	−0.05 *	0.21 **	0.21 **	−0.04	−0.22 **	0.25 **	−0.07
7. IN.R	-	-	-	-	-	-	-	0.83 **	0.74 **	0.20 **	−0.21 **	0.54 **	0.14 **	0.01	0.93 **
8 INTE.R	-	-	-	-	-	-	-	-	0.73 **	0.24 **	−0.22 **	0.60 **	0.16 **	0.04	0.93 **
9. IDEN.R	-	-	-	-	-	-	-	-	-	0.31 **	−0.13	0.51 **	0.12 **	0.13 **	0.88 **
10. INTRO.R	-	-	-	-	-	-	-	-	-	-	0.35 **	0.22 **	−0.03	0.86 **	0.27 **
11. EXT.R	-	-	-	-	-	-	-	-	-	-	-	−0.10 *	−0.17 **	0.77 **	−0.21 **
12. P.A	-	-	-	-	-	-	-	-	-	-	-	-	0.07	0.09 *	0.60 **
13. S.W.L	-	-	-	-	-	-	-	-	-	-	-	-	-	−0.11 *	0.15 **
14. A.M	-	-	-	-	-	-	-	-	-	-	-	-		-	0.06
15. C.M	-	-	-	-	-	-	-	-	-	-	-	-	-	-	-
M	3.8	3.3	4.3	1.8	1.6	1.9	3.8	3.5	4.0	2.1	1.6	2.5	3.7	1.8	3.8
SD	0.70	0.75	0.69	0.87	0.76	0.87	1.0	1.1	0.88	0.87	0.70	0.67	0.78	0.64	0.92
AS	−0.72	−0.13	−1.3	1.0	1.3	0.85	−0.96	−0.49	−1.2	0.56	1.3	0.14	−0.54	0.79	−0.89
K	0.88	−0.04	2.0	0.68	1.4	−0.7	0.50	−0.57	1.3	−0.24	2.0	−0.33	0.12	0.54	0.49

* *p* < 0.05. ** *p* < 0.01; C.S: competence satisfaction; A.S: autonomy satisfaction; R.S: relatedness satisfaction; T.C: frustration competence; T.A: frustration autonomy: T.R: frustration relatedness; IN.R: intrinsic regulation; INTE.R: integrated regulation; IDEN.R: identified regulation; INTRO.R: introjected regulation; EXT.R: external regulation: P.A: physical activity; S.W.L: satisfaction with life; A.M: autonomous motivation; C.M: controlled motivation.

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
