# Peer review of "Physical Activity as a Regulatory Variable between Adolescents’ Motivational Processes and Satisfaction with Life"

_ijerph, 2019, doi:10.3390/ijerph16152765_

Round 1

Reviewer 1 Report

Thank you for giving me an opportunity to review this paper. The present study aimed to examine the degree to which motivation, along with basic psychological needs and physical activity, predicts life satisfaction. Overall, the idea of the manuscript offers some insights to understand adolescents` psychological health. However, a novel is not well delimitated since several cross-section studies have been conducted within this subject. There are important discussion constraints. Specific comments are as below.   

Introduction

The authors made a well-established contextualization of the research problem and discussed an important matter in adolescence context. However, the introduction is far too long and somehow vague; e.g p. 2, lines 85-7. I would like to suggest a more concise SDT explanation since it is a well-stablished theoretical framework (p. 2, paragraphs 3-5). The introduction idea makes a lot of sense given what we know about the variables associations, however an original gap in literature was not offered to readers, e.g. Does this study explore some of the limitations of previous studies? which are? is the proposed causal relationship well-established?What about the mediation effect of motivation?

  Measures

Participants selection was not well described. There are many missing information regarding the number of schools in the Autonomous Community of Extremadura. What intentional cluster selection refers to? 

A critical issue is regard to physical activity assessment. Given the small sample size the use of objective measures is more recommended since PA are overestimate in adolescents. Indeed, PE physical activity is a very specific behavior and analyzing in addition to LTPA increases the risk of biases. LTPA and school-time PA should be analyzed as specific domains.

Results

My main concern is with regard to the Preliminary analysis. the investigated scales constructs are well established to allow the observation of nuances within the “latente” variable. The reduction of variables a posteriori does not make much sense. the same applies for the initial model (figure 1)

Discussion

The discussion is not focused enough; it is way too short and basically appears too speculative, not always supported by the results (e.g., p. 7, lines 234-7; lines 244-7). 

In p. 8, lines 254-6 authors assumed (references 58, 59) leisure-time physical activity as precursor of life satisfaction. What about PE and school-time physical activities. Can we generalize these inferences to other physical activity behaviours?

The fundamentals that support the applicability of the study are lacking. It would be important to know studies were the lack of association also appears and the biases that accompanies those studies. The differences between boys and girls are ignored.

The role of introjected regulation on physical activity is different from external regulation, which was evidenced in the results. However, in the discussion the authors generalize the findings of the other studies by assuming controlled motivation as all-purpose construct.

The lack of objective measurement for physical activity should be mentioned as a limitation as well.

Author Response

Comments and Suggestions for Authors

Thank you for giving me an opportunity to review this paper. The present study aimed to examine the degree to which motivation, along with basic psychological needs and physical activity, predicts life satisfaction. Overall, the idea of the manuscript offers some insights to understand adolescents` psychological health. However, a novel is not well delimitated since several cross-section studies have been conducted within this subject. There are important discussion constraints. Specific comments are as below.   

Para empezar, me gustaría destacar que para que no se produzcan problemas que den lugar a confusión en el texto, he señalado los cambios en rojo en lugar de hacerlos con el control de cambios.

R = Response

Introduction

The authors made a well-established contextualization of the research problem and discussed an important matter in adolescence context. However, the introduction is far too long and somehow vague; e.g p. 2, lines 85-7. I would like to suggest a more concise SDT explanation since it is a well-stablished theoretical framework (p. 2, paragraphs 3-5). The introduction idea makes a lot of sense given what we know about the variables associations, however an original gap in literature was not offered to readers, e.g. Does this study explore some of the limitations of previous studies? which are? is the proposed causal relationship well-established?What about the mediation effect of motivation?

The authors made a      well-established contextualization of the research problem and discussed      an important matter in adolescence context. However, the introduction is      far too long and somehow vague; e.g p. 2, lines 85-7.

R:We appreciate the reviewer's improvement suggestion to reduce the length of the introduction. In this sense, we have modified, reduced and tried to improve the explanation of the SDT, making it more concise: within the contemporary work of the Self-Determination Theory, autonomous motivation (intrinsic regulation, integrated regulation, identified regulation),  is characterized by a sense of personal will, where the behavior and the activity carried out are meaningful. This occurs when a person engages in a behavior because it is perceived as consistent with the person’s own intrinsic goals and it is felt to be important or interesting. On the another hand, controlled motivation (introjected regulation, external regulation), represents behavior that emerges from feelings of pressure, punishments, feelings of shame, external rewards or approval. Finally,  amotivation is a condition that reflects a lack of positive attitude and a feeling of uselessness, and the absence of autonomous or controlled motivation that are required to persist in an activity. 

Does      this study explore some of the limitations of previous studies? which are?

R:Regarding the question of whether the limitations of previous studies are explored, the authors have added the following: (line 92-97): However, the previous studies had a series of limitations: in some cases they do not value physical activity since research takes place in a sports or school context (physical education)[20,39,41], also some of them do not use a complete conception of the Self-Determination Theory [24,25], or they are based on a different conceptual framework to test satisfaction with life [20]. In this sense, this article offers a more complete and comprehensive view of the SDT as a theoretical framework of a structural equations modeling, in relation to the perception of physical activity levels and its effect on life satisfaction.

is the      proposed causal relationship well-established?

R: We believe that the objective of this research is well established, since the purpose of the research is to assess the extent to which thwarting or satisfaction of psychological needs will have an autonomous or controlled motivation, which will influence the level of physical activity that in the last stay will predict satisfaction with life. In addition, in order to expand the possible causal relationships, a complementary analysis has been added in which the indirect effects of the model variables have been calculated, with the aim of responding to the hypotheses postulated.

The order of enunciating the variables in the hypotheses formulated according to the model has also been changed. Therefore, it is postulated that satisfaction of basic psychological needs will be positively related to autonomous motivation, physical activity, and satisfaction with life. Likewise, the second hypothesis is that frustration of basic psychological needs will be negatively related to controlled motivation and satisfaction with life.

What      about the mediation effect of motivation?

R:  We appreciate the question and suggestion, but we did not know very well if it referred to the theoretical aspect or results. Therefore, we have added information in the theoretical section (lines 78 to 82), referring to previous studies where the value mediating motivation has been assessed.

Researches based on Self-Determination Theory have already assessed the mediating value of motivation in relation to basic psychological needs (Baena-Extremera, Gómez-López, Granero-Gallegos, & Ortiz-Camacho, 2015), quality of life (Fenner, Howie, Straker, & Hagger , 2016), physical activity, (Nurmi, Hagger, Haukkala, Araujo-Soares, & Hankonen, 2016) and subjective well-being (Orkibi & Ronen, 2017).

On the other hand, in the results section and in order to complement the relationships between variables, one more paragraph has been added with the indirect effects for each of the levels of mediation.

Measures

Participants selection      was not well described. There are many missing information regarding the      number of schools in the Autonomous Community of Extremadura. What      intentional cluster selection refers to? 

A critical issue is      regard to physical activity assessment. Given the small sample size the      use of objective measures is more recommended since PA are overestimate in      adolescents. Indeed, PE physical activity is a very specific behavior and      analyzing in addition to LTPA increases the risk of biases. LTPA and      school-time PA should be analyzed as specific domains.

R: We would      like to thank questions and suggestions made by reviewer for the improvement      of the article. Firstly, in relation to the selection of participants,      more details have been added regarding the sample and its selection      process. The paragraph included is the following one:

A total of 487 Spanish students of Secondary Education aged between 14 and 16 years (M = 15.02, SD = 0.87), 262 (53.8%) males and 225 (46.2%) females, participated in this investigation. They were from five different schools in the Region of Extremadura (Spain). The sample was selected through a hierarchical cluster sampling, considering the distance of the schools, if the teaching staff was known, the availability and the time of the researcher to travel.

R: Thanks      again for your response and suggestion for improvement. In this regard,      due to the number of individuals presented in the educational context, we      chose to assess the perception of physical activity through the Paq_A      questionnaire (Kowalski, Croker, & Donen, 2004), which was validated      in Spanish adolescents by Martínez-Gómez et al. (2009). In this sense,      during the validation process accelerometers were used giving acceptable      correlation coefficients between the accelerometer and the PAQ_A.      Likewise, the validation offered reliable reliability of 0.84. On the      other hand, recently Benitez-Porres, Alvero-Cruz, Sardinha,      López-Fernández, and Carnero (2016), used the Paq_a with accelerometers      and established cut-off points in relation to MVPA to classify the      subjects as active or inactive, following the WHO recommendations.      However, we accept that the valuation of PA is an important limitation of      our study.

Thus, in response to the reviewer's suggestion, we believe that we should change the concept of physical activity levels by perception of physical activity levels, as this may cause confusion to potential readers of the article. In relation to this, the first refers to an objective assessment of physical activity levels and the second refers to a subjective assessment. However, as previously noted, both has a significant correlation with each other.

Results

My main concern is with      regard to the Preliminary analysis. the investigated scales      constructs are well established to allow the observation of nuances within      the “latente” variable. The reduction of variables a posteriori does not      make much sense. the same applies for the initial model (figure      1)

R: We appreciate your      contribution to improve the quality of the article. The reduction of the      variables a posteriori was decided, since when making the structural      equation modeling with the Mplus 7.0 program, it gave us an error in the      output due to the excess of interactions. In this sense, it was decided to      group the variables of BPNS and BPNT, this same problem occurred when all      types of motivation were introduced, so it was decided to group and      construct the latent variables. We want to point out that there are      previous studies showing models of structural equations based on the      self-determination theory where they conducted the same constructions of      latent variables. That is to say, the hypothesized model was tested      (figure 1), but the adjustment rates obtained were not acceptable.

When considering the relations of the amotivation with the variables of the model, it was decided to delate from the model porque al ver los pesos estandarizados (standardized weights) demotivation was not significantly related to the rest of the variables of the model and, for this reason, it was decided to eliminate it and test model 2. (since it had a negative impact on the adjustment index). In this sense, a positive and a negative line was maintained where well-being was last predicted. and in some cases they eliminate the amotivation variable. In this regard, previous studies also keep out of the amotivation model or use only one of the motivational regulations to produce health benefits (Gunnel, Crocker, Mack, Wilson, & Zumbo, 2014; Kalajas-Tilga, Koka, Hein, Tilga, & Raudsepp 2019).

R: Indirect      effects have been calculated to see the relationships of the different      level variables and verify the mediating effect of the different types of      motivation and the perception of physical activity levels, if      necessary, the question asked in the introduction about the mediating      value of motivation.

Thus, the following information has been added in the manuscript within the data analysis section:

To test the indirect effects, the path model was re-estimated using bootstrapping resampling procedures (N = 5.000) to compute 95% Bias-corrected Confidence Intervals (95% BcCI) (Preacher& Hayes, 2008). If the 95% BcCI did not included zero, the indirect association was deemed significant. This model was estimated using the Maximum Likelihood (ML) estimator since MLR with bootstrapping is not yet available in Mplus.

Moreover, in the results section, the following information has been included:

The indirect effects between the variables showed: Firstly, need satisfaction was positively and significantly related (p = .016) to the PA levels via types of motivations (β = .285, 95% BcCI = [.090, .480]), whereas need thwarting was not significantly (p> .05) associated with PA levels via types of motivation (β = .088, 95% BcCI = [-.104, .281]). Secondly, need satisfaction was positively but not significantly (p> .05) related to satisfaction with life (β = .027, 95% BcCI = [-.003, .057]), via types of motivation and PA levels. In addition, need thwarting was not also significantly (p> .05) associated with satisfaction with life (β = .008, 95% BcCI = [-.013, .029]) via types of motivation and PA levels. Finally, autonomous motivation was significant indirectly (p = .032) related to satisfaction with life via PA levels (β = .057, 95% BcCI = [.013, .101]), and controlled motivation was not significantly (p> .05) associated with satisfaction with life (β = .008, 95% BcCI = [-.002, .018]).

Discussion

The discussion is not focused enough; it is way too short and basically appears too speculative, not always supported by the results (e.g., p. 7, lines 234-7; lines 244-7). 

In p. 8, lines 254-6 authors assumed (references 58, 59) leisure-time physical activity as precursor of life satisfaction. What about PE and school-time physical activities. Can we generalize these inferences to other physical activity behaviours?

The fundamentals that support the applicability of the study are lacking. It would be important to know studies were the lack of association also appears and the biases that accompanies those studies. The differences between boys and girls are ignored.

The role of introjected regulation on physical activity is different from external regulation, which was evidenced in the results. However, in the discussion the authors generalize the findings of the other studies by assuming controlled motivation as all-purpose construct.

The lack of objective measurement for physical activity should be mentioned as a limitation as well.

The discussion is not      focused enough; it is way too short and basically appears too speculative,      not always supported by the results (e.g., p. 7, lines 234-7; lines      244-7).

R: Again, the authors      appreciate the reviewer's contribution. In this sense, the discussion has      been widely modified in response to these comments. No speculations have      been included, but based on these results produced by correlations and      indirect effects, the results have been commented and compared with those      of other studies.

Therefore, the following sentence has been complemented with the information highlighted in red.

Previous studies also confirmed this positive association between the levels of more autonomous motivation and the satisfaction of the basic psychological needs many other domains (e.g., school, sport, and work). Satisfaction of the basic psychological needs and autonomous motivation were also positively associated with physical activity and satisfaction with life [37,38]

In p. 8, lines 254-6      authors assumed (references 58, 59) leisure-time physical activity as      precursor of life satisfaction. What about PE and school-time physical      activities. Can we generalize these inferences to other physical activity      behaviours?

2.  R:  The authors welcome suggestions for improvement, as they contribute to the improvement of the quality of the manuscript. In relation to the questions that refer to physical activity in the text the following has been added:

Therefore, people who feel competent at the task they perform within their peer group will present greater satisfaction with life in all its aspects. Regarding the relationship between physical activity and autonomous motivation [58], state that people with autonomous motives are more likely to engage in activities that require some effort.

In this sense, [59] emphasized that autonomous motivation for adolescents should be fostered through support for autonomy, offering alternative activities that have a rational meaning (autonomy), supporting their confidence and skills, recognizing their efforts (competence), and encouraging positive interaction with their peers (relatedness). Thus, previous studies developed in the school context showed how the motivational climate of the physical education teacher influences when developing healthy living habits [60,61]. According to the SDT, people can experience deep well-being when their social environment supports their innate needs for competence, relatedness and, above all, autonomy.

The fundamentals that      support the applicability of the study are lacking. It would be important      to know studies were the lack of association also appears and the biases      that accompanies those studies. The differences between boys and girls are      ignored.

The role of introjected regulation on physical activity is different from external regulation, which was evidenced in the results. However, in the discussion the authors generalize the findings of the other studies by assuming controlled motivation as all-purpose construct.

3.R: Again, we appreciate the recommendations suggested, and wehope that the changes conducted will be in line with the reviewer indication. First of all, we would like to respond to you the issue regarding gender. The differences in gender were a complementary finding of the objective of the study, not the main finding by which we highlight the importance of this article. That is to say, when checking a structural equations modeling, it is interesting to check the invariance with respect to gender. In this case the assumption of invariance was not fulfilled and it was added where the differences occurred. The invariance of the model in relation to gender was that this model was a variant, and that for the male gender physical activity was a more important determinant of satisfaction with life than for the female gender. However, if the reviewer considers that we should delve deeper into the differences and in some ways modify the objectives, the authors will agree to extent this issue.

In relation to controlled motivation and with regard to the relationship between introjected regulation and physical activity, the theory of self-determination states the following: “Introjected and external regulations are considered two relativel y controlled forms of motivation”. Likewise, this does not mean that introjected regulation must necessarily have a negative relationship with physical activity. In this sense, when analysed separately, introjected regulation more frequently shows a positive cross-sectional association with physical activity whereas external regulation is more commonly negatively associated with physical activity (e.g., Edmunds, Ntoumanis, & Duda, 2006; Wilson, Rodgers, & Fraser, 2002). Nevertheless, cross-sectional research has indicated that introjected regulation may be associated with higher levels of physical activity (e.g., Brunet & Sabiston, 2011; Edmunds et al., 2006). This positive relationship between introjected regulation and physical activity can be explained through the current times where the trends and fashions that surround us, exert a direct or indirect social pressure that can have an impact on a greater performance of physical activity, although I do not do it for the mere fact of enjoying but that it is carried out by feeling guilty or social acceptance.

The      lack of objective measurement for physical activity should be mentioned as      a limitation as well

4. R: We appreciate the reviewer's contribution. In this regard, in line 309-310 a limitation has been added: “Another limitation is the non-inclusion of an objective instrument for the assessment of physical activity levels”.

Reviewer 2 Report

The article deals with important and interesting issues. I consider the article positive, but I propose to introduce minor additions. 

In the introduction, the research gap should be clearly indicated.

In the methodological chapter it is necessary to describe in detail how the sample was selected for the research.

Author Response

Comments and SuggestionsforAuthors

The article deals with important and interesting issues. I consider the article positive, but I propose to introduce minor additions. 

In the introduction, the research gap should be clearly indicated.

In the methodological chapter it is necessary to describe in detail how the sample was selected for the research.

Respond to reviewer 2.

Para empezar, me gustaría destacar que para que no se produzcan problemas que den lugar a confusión en el texto, he señalado los cambios en rojo en lugar de hacerlos con el control de cambios.

R = Response

1.      Does the introduction provide sufficient background and include all relevant references?

In regards with the background the reviewer recomend that the introduction, the research gap should be clearly indicated

R: We appreciate the suggestion of improvement of the reviewer, since it contributes to the improvement of the quality of the article. In this sense, some modifications have been made in the text, with the aim to enhace the knowledge about the relationship between motivation and physical activity in the Physical Education context. Therefore, more text has been added in some of the sections of the work.

First, the theoretical mediating value of the motivation was added, to improve understanding of the relationships between variables.

Researches based on Self-Determination Theory have already assessed the mediating value of motivation in relation to basic psychological needs (Baena-Extremera, Gómez-López, Granero-Gallegos, & Ortiz-Camacho, 2015), quality of life (Fenner, Howie, Straker, & Hagger , 2016), physical activity, (Nurmi, Hagger, Haukkala, Araujo-Soares, & Hankonen, 2016) and subjective well-being (Orkibi & Ronen, 2017).

Secondly and in response to the reviewer's suggestion, the following information has been added, pointing out the limitations of previous studies that have led to the performance of this article:

However, previous studies had a series of limitations: in some cases they do not value physical activity since research takes place in a sports or school context (physical education) [20,39,41], also some of them do not use a complete conception of the Self-Determination Theory [24,25], or they are based on a different conceptual framework to test satisfaction with life [20].

Antecedentes suficientes y referencias relevantes, se recomienda clarificar el vacio de investigación.

2.      In the methodological chapter it is necessary to describe in detail how the simple was selected for the research.

R: In the methodological section, new content has been added to respnse the requirements of the reviewers.

First, it has been detailed how the sample selection process was:

A total of 487 Spanish students of Secondary Education aged between 14 and 16 years (M = 15.02, SD = 0.87), 262 (53.8%) males and 225 (46.2%) females, participated in this investigation. They were from five different schools in the Region of Extremadura (Spain). The sample was selected through a hierarchical cluster sampling, considering the distance of the schools, if the teaching staff was known, the availability and the time of the researcher to travel.

And subsequently we have added the indirect effects of motivation with the study variables to complement the structural equation model:

Data analisys

To test the indirect effects, the path model was re-estimated using bootstrapping resampling procedures (N = 5.000) to compute 95% Bias-corrected Confidence Intervals (95% BcCI) (Preacher & Hayes, 2008). If the 95% BcCI did not included zero, the indirect association was deemed significant. This model was estimated using the Maximum Likelihood (ML) estimator since MLR with bootstrapping is not yet available in Mplus.

Results

The indirect effects between the variables showed firstly need satisfaction was positively and significantlyrelated (p = .016) to the PA levels via types of motivations (β = .285, 95% BcCI = [.090, .480]), whereas need frustration was not significantly(p> .05) associated withPA levels via types of motivation (β = .088, 95% BcCI = [-.104, .281]). Secondly, need satisfaction was positively but not significantly(p> .05) related to satisfaction with life (β = .027, 95% BcCI = [-.003, .057]), via types of motivation and PA levels. In addition, need thwarting was not also significantly(p> .05) associated with satisfaction with life (β = .008, 95% BcCI = [-.013, .029]) via types of motivation and PA levels.Finally, autonomous motivation was significant indirectly (p = .032)related to satisfaction with life via PA levels (β = .057, 95% BcCI = [.013, .101]), and controlled motivation was not significantly (p> .05) associated with satisfaction with life (β = .008, 95% BcCI = [-.002, .018]).

Discussion

Finally, an attempt has been made to improve the quality of the discussion and a limitation has been added:

Another limitation is the non-inclusion of an objective instrument for the assessment of physical activity levels.

Round 2

Reviewer 1 Report

I really appreciate the effort to improve the manuscript.

Best regards